# K-Means Clustering for Shock Classification in Pediatric Intensive Care Units

**DOI:** 10.3390/diagnostics12081932

**Published:** 2022-08-10

**Authors:** María Rollán-Martínez-Herrera, Jon Kerexeta-Sarriegi, Javier Gil-Antón, Javier Pilar-Orive, Iván Macía-Oliver

**Affiliations:** 1Cruces University Hospital, 48903 Barakaldo, Spain; 2Biocruces Bizkaia Health Research Institute, 48903 Barakaldo, Spain; 3Vicomtech Foundation, Basque Research and Technology Alliance (BRTA), Mikeletegi 57, 20009 Donostia, Spain; 4Biodonostia Health Research Institute, 20018 Donostia, Spain; 5Computational Intelligence Group, Computer Science Faculty, University of the Basque Country, UPV/EHU, 48940 Donostia, Spain

**Keywords:** shock, pediatric, unsupervised learning, k-means, stratification

## Abstract

Shock is described as an inadequate oxygen supply to the tissues and can be classified in multiple ways. In clinical practice still, old methods are used to discriminate these shock types. This article proposes the application of unsupervised classification methods for the stratification of these patients in order to treat them more appropriately. With a cohort of 90 patients admitted in pediatric intensive care units (PICU), the k-means algorithm was applied in the first 24 h data since admission (physiological and analytical variables and the need for devices), obtaining three main groups. Significant differences were found in variables used (e.g., mean diastolic arterial pressure *p* < 0.001, age *p* < 0.001) and not used for training (e.g., EtCO2 min *p* < 0.001, Troponin max *p* < 0.01), discharge diagnosis (*p* < 0.001) and outcomes (*p* < 0.05). Clustering classification equaled classical classification in its association with LOS (*p* = 0.01) and surpassed it in its association with mortality (*p* < 0.04 vs. *p* = 0.16). We have been able to classify shocked pediatric patients with higher outcome correlation than the clinical traditional method. These results support the utility of unsupervised learning algorithms for patient classification in PICU.

## 1. Introduction

Shock is described as an inadequate oxygen supply to the tissues and can be classified in multiple ways. There are four classic pathophysiological variants: cardiogenic, hypovolemic, distributive, and obstructive, which explain the hemodynamic situation of the patient and are therefore useful for establishing therapy. Shock can also be classified according to its etiology: septic, anaphylactic, hemorrhagic, etc. The problem with these classifications is they are often confusing, since each pathophysiological type has different etiological varieties and each of the etiological varieties may correspond to different pathophysiological types [1,2,3,4,5]. 

When faced with a new case of shock, it is necessary to simultaneously evaluate multiple physiological variables to determine the pathophysiological variety, in order to establish a targeted therapy [3]. However, despite the time that has been spent researching hemodynamic monitoring, there are still no adequate techniques to make an accurate shock classification [4,5,6], which negatively affects the clinical management. 

Multiple studies have been performed, and classical statistical analysis has been used to determine the type of shock in each patient. However, no single solution to this historical problem has yet been found. Currently, there are multiple models of biological data analysis using artificial intelligence techniques based on machine learning [7]. The use of supervised learning for patient categorization would imply the assumption that the ground truth of the disease classification is already perfect, but in practical terms, this classification may not be optimal or be outdated. Therefore, in clinical practice, similarities between patients are continually encountered, which in the long run sometimes end up defining new categories.

The present study proposes that the use of unsupervised learning techniques based on the multiple variables routinely collected in pediatric intensive care units (PICU) could help to find patterns that clinicians may have overlooked, and that may define new classifications of types of shock in the pediatric setting. The use of clustering algorithms for the classification of shock could not only be applied globally but could also be used locally. It could be useful to study the characteristics of each hospital and classify patients into clinical groups to study treatment and evolution, similarly to that performed at the level of microbiological flora by analyzing local behavior and resistances.

## 2. Materials and Methods

A retrospective observational study was carried out for the development of a computational model for the classification of types of shock in the PICU of Cruces University Hospital. Clinical and analytical data were collected for all pediatric patients (0 to 14 years) diagnosed with any type of shock since the implementation of “IntelliSpace Critical Care and Anesthesia”, in 2012. 

Data were collected in a computerized manner through the program’s database: 180 patients had been diagnosed with shock, but only 100 had data. Hourly data of physiological, gasometric, and analytical variables were obtained. In addition, devices, age, weight, length of stay, diagnostic at discharge, and discharge reason, were recorded. 

From all the information available at the beginning, a first filtering was made by removing those variables with more than 80% of missing values and those with no clinical relevance (such as the number of central venous catheter lumens, and urinary catheter size, etc.). Then, patients with missing values in the heart rate, respiratory rate, and pressure columns were eliminated. Afterward, the resulting dataset comprised 90 patients. In order to achieve higher clinical value in the classification, only data from the first 24 h of admission were selected, and data preceding death in less than 48 h were eliminated. 

For those variables that depend on age, the values were adjusted with the z-score for that age (weight [8,9], blood pressure (using the p. 50 for height) [10], and heart and respiratory rate [11]).

After that, the mean, minimum (min), and maximum (max) values of each monitored numerical variable were calculated. A classification of the patients was performed by clustering. 

The following statistical techniques were used. Lilliefors test was used to analyze normality. Normal variables were described by mean, non-normal variables by the median, and qualitative variables by proportion. Dispersion measures have been estimated by the confidence interval of 95%. For quantitative variables with 2 groups, the *t*-test (with Welch correction if homoscedasticity was not met) or Wilcoxon test were used depending on normality. In quantitative variables with more than 2 groups, ANOVA or Kruskal–Wallis were used depending on normality, and post hoc analyses were performed with Tukey and the Mann–Whitney test with significance correction, respectively. Chi-square was used for qualitative variables. In the case of length of stay, Log-Rank was also used, with post hoc analysis using Bonferroni correction. A *p* < 0.05 was considered the cut-off point for statistical significance. 

K-means was the algorithm selected for clustering [12]. This algorithm classifies individuals into multiple groups so that individuals within the same group are as similar as possible, while individuals from different groups are as different as possible. This similarity is estimated using the Euclidean distance [13], so that if two patients have very different values for a variable, the distance will be high, and vice versa. Each group is represented by its centroid, which corresponds to the mean of all individuals in that group, the initial centroids are randomly selected by the algorithm, and this may influence the results, which is why 25 initial configurations were attempted. As a characteristic of the k-means algorithm, it is necessary to define both the grouping variables and the number of clusters. 

For the clustering classification, only the relevant physiological variables with less than 45% of missing values were used. To study the optimal number of clusters, the elbow method and the average silhouette method were used, and several tests were performed to check the spatial distribution of the groups and the number of patients per group. Three was taken as the optimal number of clusters. 

Once the grouping variables and the number of clusters were selected, the data were prepared: each missing value was estimated by averaging the remaining individuals, and then the data were standardized with z-score [14] to make all variables comparable. 

Although it has been shown that good results can be achieved by giving more weight to some variables than others [15], in this study, all variables have been treated with equal importance to avoid both clinician subjectivity and overfitting when assigning weights (due to the size of the database). 

Once the clusters were selected, the clinical significance of those clusters was sought as follows: The characteristics of each group were studied to determine whether there were differences between them.The correlation between the unsupervised classification and the discharge diagnosis was studied.It was assessed whether the classification was related to the outcomes (mortality and length of stay).It was tested whether the new classification had a greater association with outcomes than the classic classification.

All analyses were performed with R-Studio; tables and graphs were made with R-studio and Microsoft Excel. 

## 3. Results

K-means classification was performed using the variables indicated in Table A1 (shown in the appendix due to its size). The following clusters were obtained: 46 patients, 18 patients, and 26 patients (Figure 1). 

### 3.1. Analysis of Variables Used for Clustering

Each of the variables was compared between the different clusters (Table A1). Among all the variables, the need for ECMO was the one with the greatest weight, so the clustering was performed again without it. The clusters obtained were 45, 19 and 26 patients, almost the same distribution as before (ECMO: 0%, 79% and 2%, respectively, *p* < 0.001).

Cluster 1 was the cluster with the highest proportion of female patients (56%), the one with the lowest weight for age, and the one with the lowest mean arterial pressure and diastolic pressure. The patients in this cluster had the lowest carboxyhemoglobin, the highest calcium ion, the lowest creatinine, and the highest number of neutrophils. The cluster with the highest proportion of intracranial catheters and thermal blankets and the lowest proportion of patients with hemodiafiltration. 

Cluster 2 was characterized as the one with lower mean age, heart and respiratory rate, systolic blood pressure, oxygen saturation, venous oxygen saturation, temperature, and higher inspired oxygen fraction. It was the group of patients with higher daily diuresis, wider capillary glycemia range, lower calcium ion levels, higher phosphate levels, lower C-reactive protein levels, higher lymphocytosis and lower neutrophilia. It was the cluster with the highest proportion of patients with ECMO, mechanical ventilation and hemodiafiltration.

Cluster 3 was the one with older patients and higher weight for age. These patients had the highest heart rate and respiratory rate, highest blood pressure, lowest daily diuresis, highest oxygen saturation and lowest inspired oxygen fraction, highest temperature, venous oxygen saturation and carboxyhemoglobin. This was the group with higher capillary glycemia, creatinine, lower phosphate levels, higher C-reactive protein, neutropenia and lymphopenia. In addition, this group had the lowest proportion of patients on mechanical ventilation, ECMO and heat blanket. 

Clusters 1 and 3 were the most similar; however, cluster 3 presented higher age (*p* < 0.001), higher weight for age (*p* < 0.006) and higher proportion of males (*p* < 0.04). It presented higher levels in the average, maximum and minimum values of all tensions (*p* < 0.001). Finally, cluster 3 presented lower lymphocytosis and neutrophilia (*p* < 0.001) and higher lymphopenia and neutropenia (*p* < 0.001), in addition to a higher C-reactive protein (*p* < 0.001).

### 3.2. Analysis of Variables Not Used for Clustering

Each variable was compared among the three clusters (Table A1). Exhaled CO2 pressure showed important differences, with maximum levels in cluster 1, followed by cluster 3; cluster 2 presented much lower levels than the other two. Troponin also presented significant differences, with the highest level in cluster 2, followed by cluster 1 and with minimum levels in cluster 3.

### 3.3. Relationship between Clustering and the Classic Classification

Cardiogenic shock presented different proportions between groups 1 and 2 (*p* < 0.001) and 2 and 3 (*p* < 0.001); however, groups 1 and 3 did not present differences. Inflammatory shock presented significant differences between all combinations (1 vs. 2, *p* < 0.03; 2 vs. 3, *p* < 0.001; 3 vs. 1, *p* < 0.04). Hypovolemic shock was not specifically associated with any group. Cluster 2 had the highest number of postoperative cardiac surgery patients (44%, *p* < 0.001), while cluster 3 had the highest number of oncologic patients (31%, *p* = 0.003). In Figure 2 the absolute number of patients per classic classification of each cluster is shown.

### 3.4. Analysis of Outcomes According to Clustering

Both lengths of stay and death were chosen as outcomes. In the case of length of stay (Figure 3, part A), significant differences were found between the three groups (*p* < 0.03), mainly because of the differences between groups 2 and 3 (*p* < 0.02). Groups 1 and 2 did not present significant differences (*p* < 0.11), and neither did groups 1 and 3 (*p* < 0.28). Using Log-Rank, differences were also found between the three groups (*p* = 0.01), mainly due to groups 2 and 3 (*p* < 0.02); between groups 1 and 2, no differences were found (*p* = 0.18), nor between groups 1 and 3 (*p* = 0.8). Significant differences were observed between clusters in survival (*p* < 0.04), also mainly due to differences with cluster 2. Between clusters 1 and 2, there were significant differences (*p* < 0.02), and the same was true between clusters 2 and 3 (*p* < 0.03). However, between groups 1 and 3, there were no differences (*p* = 1).

### 3.5. Prediction of Outcomes by Classic Classification

Patients with cardiogenic shock had significantly longer length of stay than patients without cardiogenic shock (Wilcoxon *p*-value < 0.02, Log-Rank *p*-value = 0.01). Patients with inflammatory shock had shorter length of stay than patients without it (Wilcoxon *p*-value < 0.02, Log-Rank *p*-value = 0.05). As for patients with hypovolemic shock, there was no significant association with length of stay (Figure 3, part B). 

Cardiogenic and inflammatory shock presented an inverse association between them, only one patient presented both types of shock, and none presented either of them (*p* < 0.001). Given this circumstance and the fact that it makes no sense for either type of shock to shorten the length of stay, it is most likely that the true determinant of length of stay is cardiogenic shock.

As for the probability of death, none of the three classic types of shock was associated with the survival rate (Table 1).

Cardiac postoperative patients had significantly longer length of stay than non-postoperative patients (9 vs. 5 days, Wilcoxon *p*-value < 0.03, Log-Rank *p*-value = 0.01), while oncologic patients did not show significant differences in length of stay with respect to non-oncologic patients (3 vs. 6 days, Wilcoxon *p*-value = 0.15, Log-Rank *p*-value = 0.1). Regarding mortality, no significant differences were found between groups in any case (cardiac surgery: 8% vs. non-cardiac surgery: 12%, *p* = 1; oncologic: 0% vs. non-oncologic: 13%, *p* = 0.45).

Globally, no significant differences were found in the LOS between patients that died and patients that survived (4.5 vs. 6 days, Wilcoxon *p*-value = 0.91, Log-Rank *p*-value = 0.8). However, analyzing cardiogenic shock patients, it was found that patients that died had a shorter LOS than survivors (4 vs. 13 days, Wilcoxon *p*-value = 0.04, Log-Rank *p*-value = 0.004). Additionally, cluster 2 showed this behavior (4 vs. 21.5 days, Wilcoxon *p*-value = 0.02, Log-Rank *p*-value = 0.006). The other two types of shock and clusters 1 and 3 presented the opposite trend but with no significant differences. 

## 4. Discussion

It is important to consider that the clustering classification has been performed with data of a specific hospital so that it is not generalizable. Post-operative care for congenital heart disease, the absence of cardiac transplantation or the presence of a pediatric oncology unit are some of the particularities that determine the type of patients admitted with shock in this hospital, and therefore the classification made. Therefore, the proposal in this article is not the generalization of the model presented, but the demonstration that clustering algorithms (specifically k-means) can be useful for classifying shocked pediatric patients, and that they can even be more accurate than clinical classification. 

Data available were mainly from two groups according to the classical classification, patients with cardiogenic shock and patients with inflammatory shock. However, clustering classification was made in three groups, and it seems optimal in its two-dimensional representation (Figure 1 and Figure 2). 

The analysis of the different classificatory variables showed that some were more relevant than others. Specifically, ECMO was the variable with the greatest discriminatory weight, and therefore, a new analysis was performed after eliminating this variable. When this variable was eliminated, the groups barely changed, which ruled out the possibility that the good clustering results were due to this variable, and demonstrated the classificatory quality of the algorithm that could discriminate ECMO patients.

Previous studies have shown that clustering algorithms tend to group intensive care patients into physiologically similar groups [16]. Cluster 1 appeared to belong to a group of low-weight, hypotensive infants and young children, mainly with septic shock and some cardiogenic shock. Cluster 2 seemed to correspond to the most severe patients, infants with cardiogenic shock, in a large proportion of postoperative cardiac patients (44%, *p* < 0.001). Cluster 3 seemed to correspond to the least severe patients, neutropenic children, with apparent distributive septic shock, in a large proportion of oncologic patients (39%, *p* < 0.004). 

The two main classical types of shock were associated with different clusters; cluster 2 had the highest proportion of cardiogenic shock (72%, *p* < 0.001) and inflammatory shock was mainly associated with clusters 1 (52%) and 3 (81%), but with significant differences between them (*p* < 0.04) and with differences between them and cluster 2 (*p* < 0.03). Considering that the clustering classification was performed only using data from the first 24 h, and the classical classification was performed based on the discharge diagnosis, these results are very promising, and they could suggest that clustering is able to discriminate patients earlier than the clinical eye. 

As for the assessment of whether this new classification was able to predict outcomes, the length of stay was different among the three clusters (11 vs. 4 days, Log-Rank *p* < 0.02). Graphically (Figure 3), the Kaplan–Meier curves were different for each cluster of patients; therefore, the absence of differences between groups 1–2 and 2–3 could be due to the small amount of data. The prediction of mortality was also satisfactory, since groups 1 and 2 (7% vs. 33%, *p* < 0.02) and 2 and 3 (33% vs. 4%, *p* < 0.03) presented significant differences; groups 1 and 3, both having such a small proportion of deceased patients, did not present differences. These results are consistent with those of previous analyses, which also suggest that clustering can provide prognostic information [17].

The final objective of this study was to determine whether the use of clustering algorithms could be better than the use of clinical classifications. To do so, it was necessary to compare the outcome prediction ability of the two types of classifications, but is important to bear in mind that the clustering classification was performed with data from the first 24 h; moreover, clustering classification has fewer patients per group than the classical classification, so differences between groups were less likely to be significant.

Cardiogenic shock was associated with a longer length of stay (9 vs. 5 days, Log-Rank *p* = 0.01) with the same significance as cluster classification (6 vs. 11 vs. 4 days, Log-Rank *p* = 0.01); therefore, in this scenario, clustering did not improve the prediction of length of stay. The association with mortality by classical categorization was not significant in any case (cardiogenic shock 21% vs. non-cardiogenic shock 8%, *p* = 0.16); however, the association with mortality using clustering classification was significant (7% vs. 33% vs. 4%, *p* = 0.003); therefore, classification by clustering improved the prediction of mortality.

Although the field of artificial intelligence applied to medicine has been booming in recent years, most of the work focuses on the technical side of the algorithms. The results of the present study represent an approximation between the technical world and the clinical world. It seems that, even with such a small sample, the presented method is able to classify patients with a higher outcome correlation than the clinical method. Other studies have also presented positive results in the application of this type of clustering in other pathologies [17].

The capacity for early classification of shocked patients presents multiple clinical utilities. From the evolutionary point of view, a correct classification allows the clinician to identify the type of patient he/she is dealing with and, based on the historical behavior of similar patients, allows him/her to predict the evolution of the patient. From the therapeutic point of view, patients who behave the same physiologically tend to respond similarly to the same therapies, so a correct classification allows for improved treatment.

The classification obtained in the present study is not generalizable, but what is generalizable is the method applied to obtain it. However, it is necessary to consider some limitations. When data are collected digitally, sometimes they present incorrect recordings, which has to be cautiously interpreted [17,18,19], but this limitation is also present in clinical classification. Moreover, k-means algorithm requires pre-specifying a number of clusters, which means it is not absolutely automatic, it also needs the imputation of missing values [17] and it is very sensitive to outliers. Additionally, the size of the trained dataset is small and from a single hospital, so the results, pending validation, are not generalizable to other hospitals. Finally, the variables that have been used are static, i.e., they have been atemporalized, and thus, it would be interesting, as future work, to carry out this experiment but considering all the sequence of the data.

Despite these issues, the present results are still encouraging, and they support the utility of these methods in the classification of patients with shock in pediatric intensive care, something that could help to implement guided therapies [17]. If each hospital develops its own classification, which will be different according to their characteristics, or if a multicenter classification is performed, this may help to improve the clinical management of pediatric patients with shock.

## 5. Conclusions

In conclusion, the present study demonstrates the capacity of the k-means algorithm to correctly classify pediatric patients with shock and shows a promising future for the use of unsupervised machine learning techniques in pediatric critical care, something that could lead to an improvement in clinical management. However, further studies are needed to validate this method on a larger scale.

## Figures and Tables

**Figure 1 diagnostics-12-01932-f001:**
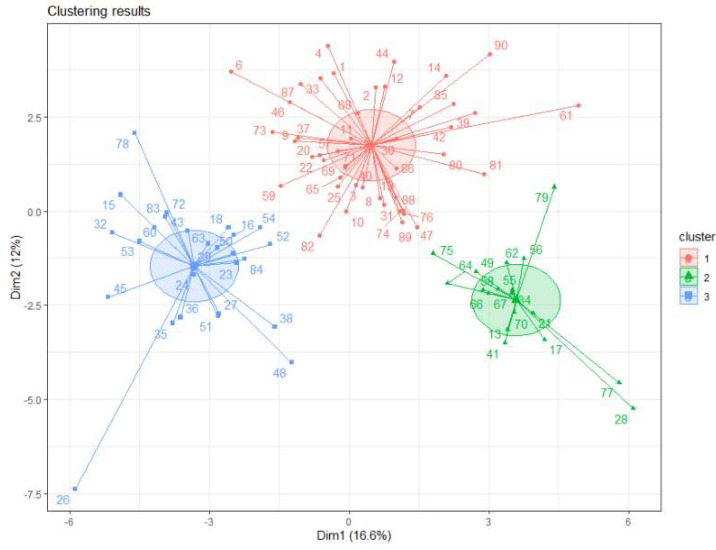
Two—dimensional representation of the clusters based on principal component analyses.

**Figure 2 diagnostics-12-01932-f002:**
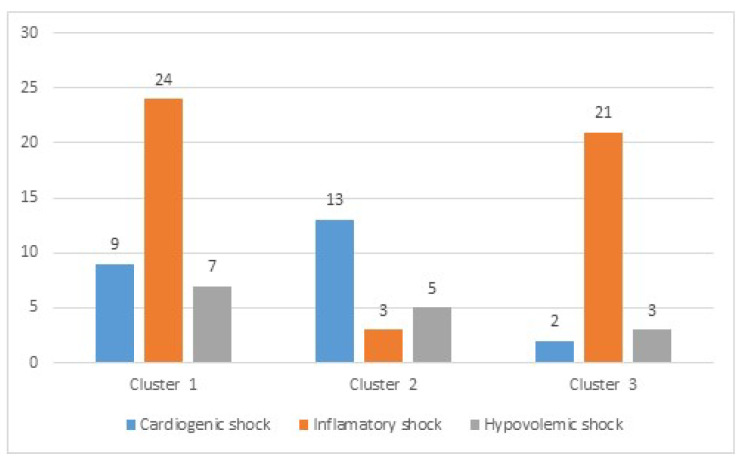
Number of patients with each type of shock in every cluster.

**Figure 3 diagnostics-12-01932-f003:**
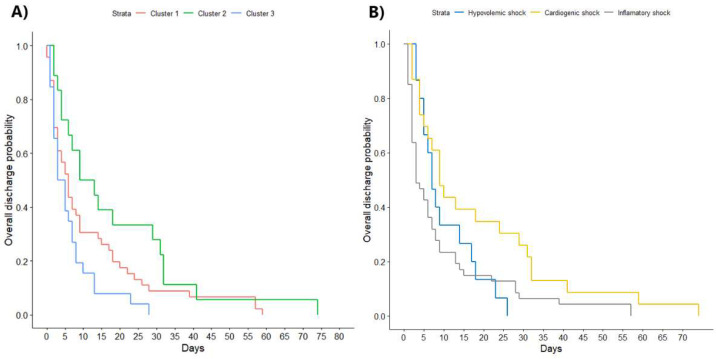
Kaplan–Meier curves for the length of stay in the intensive care unit (**A**) according to clusters and (**B**) according to classic classification.

**Table 1 diagnostics-12-01932-t001:** Association between each type of shock and both length of stay and death.

**Types of Shock**	**No Cardiogenic**	**Cardiogenic**	**Wilcoxon/χ2 (*p* Value)**	**Log-Rank (*p* Value)**
**Median length of stay (days)**	5 (3; 7)	9 (4; 24)	0.02	0.01
**Exitus**	0.08 (0.01; 0.14)	0.21 (0.03; 0.38)	0.16	-
	**No inflammatory**	**Inflammatory**	**Wilcoxon/χ2 (*p* value)**	**Log-Rank (*p* value)**
**Median length of stay (days)**	7.5 (5; 13)	3 (2; 6)	0.01	0.05
**Exitus**	0.14 (0.03; 0.25)	0.08 (0; 0.16)	0.58	-
	**No hypovolemic**	**Hypovolemic**	**Wilcoxon/χ2 (*p* value)**	**Log-Rank (*p* value)**
**Median length of stay (days)**	5 (3; 7)	7 (5; 14)	0.17	1
**Exitus**	0.11 (0.04; 0.18)	0.13 (0; 0.33)	1	-

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
