# Peer review of "K-Means Clustering for Shock Classification in Pediatric Intensive Care Units"

_diagnostics, 2022, doi:10.3390/diagnostics12081932_

Round 1

Reviewer 1 Report

Congratulations for this interesting work and thank you for sending it to Diagnostics for its possible publication.

Below I enclose some comments and suggestions in case you wish to take them into account, with the hope that they could help you improve the quality of the manuscript.

Material and Methods.

Line 84. Consider adding the dispersion measures used: Mean (SD), Median (IQR), etc.

Line 86 (and throughout the text). Please consider changing the name of the code function (Wilcox.test) to the statistical name “Wilcoxon test”.

The sentences in lines 70 and 102 refer to the elimination of variables that had more than 80% or 45% missing values. Do they refer to different types of variables (preprocessing, clustering...)? This is a bit confusing. Please clarify it.

Results. 

In the text, lines 163-165, it says “Exhaled CO2 pres-163 sure showed important differences, with maximum levels in cluster 1, followed by cluster 2; cluster 3 presented much lower levels than the other two.” But in the table, the lowest values are for cluster 2. Please, correct this.

Line 174. It says “… 3 vs. 4, p < 0.04).”. Was there a 4th cluster? Or does it refer to group 1 vs. 3?

In lines 187-8, where it says “Significant differences between clusters were objectified in survival (p < 0.04)..”  Please, consider changing to “Significant differences were observed between clusters in survival (p < 0.04)…”

Line 193.: “Table 2. This is a table. Tables should be placed in the main text near to the first time they are cited”   Consider replacing this by the title of the table and a brief explanation of it.

For 95% CI, please consider semicolon (;) instead of colon (:). 

The results regarding mortality and length of stay are difficult to interpret. As a general rule, less severe patients should have a lower LOS, but it could also be the case that more severe patients die earlier. Therefore, for a more appropriate interpretation of the results, an independent analysis of the LOS of surviving and deceased patients should perhaps be carried out.

Please, review Appendix Table 1. It seems to contain some inconsistencies. For example, in the row “Inspirited oxygen fraction in % (mean)” in the columns corresponding to the clusters it says “md:” (median?). By the way, if the parameter appears in the first column, it is unnecessary to repeat it in the next three columns. In addition, the sixth column (“test”) could be eliminated, given that the test used in each case is already explained in the Methods section.

Another issue that give rise to some confusion is, for instance, when the variable measure says “mn” (mean) and the test “Chi-squared” (see the last raw “exitus”, and some others).

In some rows the 95% CI (for example “End expiratory pressure (mean)” are nor given. Please, add it or explain why.

Minor (grammar and typos).

Some examples:

Line 42. “…which affect negatively to the clinical management.” Consider “… which negatively affects the clinical management.”

Line 49. “may be not optimal”. Consider “may not be optimal”.

Line 69. “diagnostic at discharge”. Consider “discharge diagnosis”.

Etc.

Author Response

Rebuttal

Dear Editors,

We want to thank the reviewer for his/her valuable time and useful contributions. We greatly appreciate the given feedback which will definitely help improve our manuscript.

We have edited the manuscript to address their concerns. Nevertheless, please, do not hesitate to contact us for any further comments or suggestions. The lines in which we specify the changes are with the track changes extended.

Thank you again for your review and we look forward to hearing from you soon regarding our submission.

We have answered the comments one by one below:

Comment 1

REVIEWER:

( )

( )

( )

Congratulations for this interesting work and thank you for sending it to Diagnostics for its possible publication.

Below I enclose some comments and suggestions in case you wish to take them into account, with the hope that they could help you improve the quality of the manuscript.

AUTHORS:

Thank you very much for the good feedback you have given us. We have heeded the comments and we believe that it has been improved because of it. Thank you.

Comment 2

REVIEWER:

Material and Methods.

Line 84. Consider adding the dispersion measures used: Mean (SD), Median (IQR), etc.

Dispersion measures have been estimated by the confidence interval of 95%.

Line 86 (and throughout the text). Please consider changing the name of the code function (Wilcox.test) to the statistical name “Wilcoxon test”.

AUTHORS

Done, thank you.

Comment 3

REVIEWER

The sentences in lines 70 and 102 refer to the elimination of variables that had more than 80% or 45% missing values. Do they refer to different types of variables (preprocessing, clustering...)? This is a bit confusing. Please clarify it.

AUTHORS

On the one hand, some variables have been discarded from the study (80%), i.e. these variables have been considered for the general statistical analysis, on the other hand, those with less missing values than 45% have been used for clustering. We have changed it to make it easier to understand. Check lines 71-73 and 108-110.

Comment 4

REVIEWER

Results. 

In the text, lines 163-165, it says “Exhaled CO2 pres-163 sure showed important differences, with maximum levels in cluster 1, followed by cluster 2; cluster 3 presented much lower levels than the other two.” But in the table, the lowest values are for cluster 2. Please, correct this.

AUTHORS

Changed, thank you

Comment 5

REVIEWER

Line 174. It says “… 3 vs. 4, p < 0.04).”. Was there a 4th cluster? Or does it refer to group 1 vs. 3?

AUTHORS

It was a mistake, thank you

Comment 6

REVIEWER

In lines 187-8, where it says “Significant differences between clusters were objectified in survival (p < 0.04)..”  Please, consider changing to “Significant differences were observed between clusters in

survival (p < 0.04)…”

AUTHORS

Changed, thank you

Comment 7

REVIEWER

Line 193.: “Table 2. This is a table. Tables should be placed in the main text near to the first time they are cited”   Consider replacing this by the title of the table and a brief explanation of it.

AUTHORS

Changed

Comment 8

REVIEWER

For 95% CI, please consider semicolon (;) instead of colon (:). 

AUTHORS

We have changed the table according to it.

Comment 9

REVIEWER

The results regarding mortality and length of stay are difficult to interpret. As a general rule, less severe patients should have a lower LOS, but it could also be the case that more severe patients die earlier. Therefore, for a more appropriate interpretation of the results, an independent analysis of the LOS of surviving and deceased patients should perhaps be carried out.

AUTHORS

Thank you for the recommendation. We have performed the analysis you propose and we have found that just cluster 2 and cardiogenic shock presented differences in LOS related with survival. It seems that patients that die from a cardiac cause do so during the first few days (lines 221-227)

Comment 10

REVIEWER

Please, review Appendix Table 1. It seems to contain some inconsistencies. For example, in the row “Inspirited oxygen fraction in % (mean)” in the columns corresponding to the clusters it says “md:” (median?). By the way, if the parameter appears in the first column, it is unnecessary to repeat it in the next three columns. In addition, the sixth column (“test”) could be eliminated, given that the test used in each case is already explained in the Methods section.

AUTHORS

Thank you for bringing this matter to our attention because it may not have been fully explained. We have tried to explain it in a better way in material and methods. The min, mean and max of each variable is calculated over the first 24 hours. After that, each value (min, mean, max) of each variable, of each patient, is used for clustering. What we show on table 1 is the mean or median of that values (min, mean, max) for each variable.

Comment 11

REVIEWER

Another issue that give rise to some confusion is, for instance, when the variable measure says “mn” (mean) and the test “Chi-squared” (see the last raw “exitus”, and some others).

AUTHORS

Thank you for your point, we have changed it for p (proportion).

Comment 12

REVIEWER

In some rows the 95% CI (for example “End expiratory pressure (mean)” are nor given. Please, add it or explain why.

AUTHORS

Because in those cases there is very little data to estimate the confidence interval.

Comment 13

REVIEWER

Minor (grammar and typos).

Some examples:

Line 42. “…which affect negatively to the clinical management.” Consider “… which negatively affects the clinical management.”

Line 49. “may be not optimal”. Consider “may not be optimal”.

Line 69. “diagnostic at discharge”. Consider “discharge diagnosis”.

Etc.

AUTHORS

All corrected, thank you very much

Reviewer 2 Report

In this study, k-means algorithm was applied in the first 24 hours data since admission (physiological and analytical variables and the need for devices) obtaining 3 main groups of shock patients. A very rigorous statistical process was used throughout the analysis. the present study demonstrates the capacity of the k-means algorithm to correctly classify pediatric patients with shock and shows a promising future for the use of unsupervised machine learning techniques in pediatric critical care. 

Here are some of my concerns:

1.     K-means is essentially a clustering based on patient similarity or distance, so it is crucial to define the distance between patients. The method in this paper uses Z-valuation to allow comparison between different features, but with the same weight between all features. The features selected are therefore critical. I noted that the important admission diagnosis information of the patient is not included, can you explain why this important information is not used? Perhaps it is because current methods are only able to handle some quantitative data, and there are already methods that can handle patient distances in conceptual form [1]. I believe that using a more comprehensive patient similarity analysis followed by K-means clustering should yield better results. As the current clustering considers incomplete factors and reflects more of the differences between these chosen indicators, could the resulting limitations be discussed further.

[1] Jia et al. Using the distance between sets of hierarchical taxonomic clinical concepts to measure patient similarity. BMC Medical Informatics and Decision Making. 2019 19:91. 

2.     There should be further discussion on how this approach can be applied clinically. When faced with a new patient, how can the clinician effectively categorise him into a certain grouping? And what are the differences in clinical decision making for a patient categorised into a particular subgroup? 

3.     The limitations of this study should be discussed.

Some minor issues:

1.     Can you show the Kaplan-Maier curve of the classic classificiations and show their differences as Figure 3.

2.     The caption of Table2 was missed.

Author Response

Rebuttal

Dear Editors,

We want to thank the reviewer for his/her valuable time and useful contributions. We greatly appreciate the given feedback which will definitely help improve our manuscript.

We have edited the manuscript to address their concerns. Nevertheless, please, do not hesitate to contact us for any further comments or suggestions. The lines in which we specify the changes are with the track changes extended.

Thank you again for your review and we look forward to hearing from you soon regarding our submission.

We have answered the comments one by one below:

Comment 1

REVIEWER:

( )

( )

( )

Comments and Suggestions for Authors

In this study, k-means algorithm was applied in the first 24 hours data since admission (physiological and analytical variables and the need for devices) obtaining 3 main groups of shock patients. A very rigorous statistical process was used throughout the analysis. the present study demonstrates the capacity of the k-means algorithm to correctly classify pediatric patients with shock and shows a promising future for the use of unsupervised machine learning techniques in pediatric critical care. 

Here are some of my concerns:

  1. K-means is essentially a clustering based on patient similarity or distance, so it is crucial to define the distance between patients. The method in this paper uses Z-valuation to allow comparison between different features, but with the same weight between all features. The features selected are therefore critical. I noted that the important admission diagnosis information of the patient is not included, can you explain why this important information is not used? Perhaps it is because current methods are only able to handle some quantitative data, and there are already methods that can handle patient distances in conceptual form [1]. I believe that using a more comprehensive patient similarity analysis followed by K-means clustering should yield better results. As the current clustering considers incomplete factors and reflects more of the differences between these chosen indicators, could the resulting limitations be discussed further.

[1] Jia et al. Using the distance between sets of hierarchical taxonomic clinical concepts to measure patient similarity. BMC Medical Informatics and Decision Making. 2019 19:91. 

AUTHORS

First of all, thank you for bringing up this issue. On the one hand, we have preferred not to include diagnoses because the objective of the study was to do the clustering without considering the current methods of diagnosis, we wanted to rely exclusively on clinical data, and the inclusion of the current diagnoses would strongly influence the clusters. On the other hand, we have not added weights to the variables when estimating distances because (1) we want to avoid the subjectivity of the clinicians (biased by the current diagnostic methods), and (2), doing it by trial and error or some automatic mathematical method can cause significant overfitting, especially considering the size of our sample. We have explained it in lines 117-120. 

Comment 2

REVIEWER:

( )

( )

( )

  1. There should be further discussion on how this approach can be applied clinically. When faced with a new patient, how can the clinician effectively categorise him into a certain grouping? And what are the differences in clinical decision making for a patient categorised into a particular subgroup? 
AUTHORS

Thank you for the recommendation. We have added some comments at the end of the discussion and in the conclusion (lines 306-311 and 333-334)

Comment 3

REVIEWER:

( )

( )

( )

  1. The limitations of this study should be discussed.
AUTHORS

We have extended the limitations of this study in the paragraph on lines 312-323. 

Comment 4

REVIEWER:

( )

( )

( )

Some minor issues:

  1. Can you show the Kaplan-Maier curve of the classic classificiations and show their differences as Figure 3.
AUTHORS

Thank you really much for the recommendation. We have added the curve to the manuscript (line 226)

Comment 5

REVIEWER:

( )

( )

( )

  1. The caption of Table2 was missed.
AUTHORS

Corrected

Round 2

Reviewer 2 Report

Thanks to the author for his patient reply. I think the author has answered my concerns very well. There is only one small issue that can be considered if there is a chance, the reference to Figure 4 is missing in the text, and I suggest that Figure3 and Figure4 can be put together to help the reader understand the relationship between the two.